Association between MTHFR gene polymorphisms and H-type hypertension in patients with ischemic stroke

Zhou Bo 1
Yang Tingting 2
An Shicang 3
Xu Qike 4
Liang Yuna 5 891623778@qq.com
An Xiangyang 2 zxyyaxy@163.com
1 Postdoctoral Workstation, The Affiliated Taian City Central Hospital of Qingdao University , Taian, Shandong , China
2 General Obstetrics and Gynecology, The Affiliated Taian City Central Hospital of Qingdao University , Taian, Shandong , China
3 Laboratory Medicine Department, The Affiliated Taian City Central Hospital of Qingdao University , Taian, Shandong , China
4 Radiology Department, The Affiliated Taian City Central Hospital of Qingdao University , Taian, Shandong , China
5 Ultrasound Diagnosis and Treatment Department, The Affiliated Taian City Central Hospital of Qingdao University , Taian, Shandong , China
Anson Lesley
Electronic publication date: 2025 Oct 10
Publication date: 2025
Volume: 13
Electronic Location ID: e20210
Received 2025 Feb 19; Accepted 2025 Sep 18
Copyright: © 2025 Zhou et al.
Copyright year: 2025
Copyright holder: Zhou et al.
License: This is an open access article distributed under the terms of the Creative Commons Attribution License, which permits unrestricted use, distribution, reproduction and adaptation in any medium and for any purpose provided that it is properly attributed. For attribution, the original author(s), title, publication source (PeerJ) and either DOI or URL of the article must be cited.
License URL: https://creativecommons.org/licenses/by/4.0/

Keywords: MTHFR, Hypertension, Homocysteine, Ischemic stroke

Funding: Taian City Science and Technology Development Plan Project 2020NS141, 2021NS372 and 2023NS440 This work was supported by the Taian City Science and Technology Development Plan Project (2020NS141, 2021NS372, 2023NS440). The funders had no role in study design, data collection and analysis, decision to publish, or preparation of the manuscript.

==============================
Background

Methylenetetrahydrofolate reductase (MTHFR) is a key enzyme in homocysteine metabolism. Its 677C>T and 1298A>C polymorphisms can reduce enzyme activity, potentially elevating homocysteine levels. H-type hypertension (hypertension with homocysteine ≥10 μmol/L) is an important risk factor for ischemic stroke, and its synergistic effect exacerbates vascular damage. However, the association between these MTHFR polymorphisms and elevated homocysteine levels in patients with hypertension complicated by ischemic stroke remains unclear. This study aimed to investigate the association between MTHFR gene polymorphisms and H-type hypertension in patients with ischemic stroke.

Methods

A total of 215 patients with ischemic stroke and hypertension admitted to the Department of Neurology at the Taian City Central Hospital from June 2021 to December 2022 were enrolled. General clinical data and biochemical indicators were collected. MTHFR genotyping was performed using a universal sequencing kit and a TL998A fluorescence detector. Linkage disequilibrium was analyzed via SHEsis software. Statistical analyses were conducted using SPSS 25.0. P < 0.05 indicates that the difference is statistically significant.

Results

Among patients with ischemic stroke combined with hypertension in this region, the proportion of H-type hypertension was 89.3%. The proportion of males in the H-type hypertension group was significantly higher than in the non-H-type hypertension group (P < 0.05). The genotype and allele distributions of MTHFR (677C>T) (risk allele: T) differed significantly between groups (P < 0.05): the H-type group had a higher frequency of the TT genotype (47.4% vs. 17.4%) and T allele (67.2% vs. 50.0%). Multivariate logistic regression analysis showed that the MTHFR (677C>T) TT genotype was an independent risk factor for H-type hypertension (P = 0.021, OR = 2.615, 95%CI [1.154–5.926]). For haplotypes with a frequency >3%, there were three haplotypes of MTHFR (677C>T)/(1298A>C). The C-A haplotype was a protective factor for H-type hypertension (P = 0. 028, OR = 0.485, 95%CI [0.252–0.934]), while the T-A haplotype was a risk factor (P = 0.022, OR = 2.029, 95%CI [1.096–3.756]).

Conclusion

In patients with ischemic stroke, the MTHFR (677C>T) TT genotype is an independent risk factor for H-type hypertension. For haplotypes with a frequency >3%, the C-A haplotype was a protective factor for H-type hypertension, whereas the T-A haplotype was a risk factor.

Introduction

With societal development, lifestyle changes, accelerated population aging, and the increasingly prominent unhealthy lifestyles, cerebrovascular disease has become one of the main causes of death (Zhou et al., 2019). Hypertension is the most common comorbidity in hospitalized stroke patients, accounting for approximately 66.6% (National Center for Cardiovascular Diseases, China Cardiovascular Health and Disease Report Writing Group, 2024). The number of hypertensive patients in China has reached 245 million (National Center for Cardiovascular Diseases, China Cardiovascular Health and Disease Report Writing Group, 2024), with H-type hypertension accounting for about 75% (Chen et al., 2023; Qian et al., 2021). H-type hypertension is defined as essential hypertension with a homocysteine concentration ≥10 μmol/L. The risk of cardiovascular and cerebrovascular diseases in H-type hypertension patients is five times higher than that in hypertension patients (two instances of resting systolic blood pressure ≥140 mmHg and/or diastolic blood pressure ≥90 mmHg or history of hypertension) (Zhang et al., 2023a). Studies have shown that H-type hypertension can significantly increase the incidence of ischemic stroke (Yang et al., 2023; Yu et al., 2023), and lead to higher mortality (Zhao et al., 2022). As an important risk factor for ischemic stroke, H-type hypertension and ischemic stroke exacerbate vascular damage through synergistic effects, significantly increasing the risk of stroke occurrence (Chen et al., 2025). In patients with H-type hypertension, elevated homocysteine levels increase stroke risk and are associated with the methylene tetrahydrofolate reductase (MTHFR) C677T polymorphism. The CC/CT genotype shows an increased risk (HR = 3.1) (Zhao et al., 2017). Therefore, exploring the factors influencing H-type hypertension is crucial for reducing cardiovascular and cerebrovascular events.

MTHFR, located on chromosome 1p63.6, is an important enzyme in folate and homocysteine metabolism (Raghubeer & Matsha, 2021). The MTHFR (677C>T) and MTHFR (1298A>C) are the common mutation sites in MTHFR, and exhibit a synergistic effect (Zhou et al., 2025). Concurrent mutations may further reduce enzyme activity. The MTHFR (677C>T) is a mutation at nucleotide 677 from C to T, which leads to a change in the encoded protein to change from alanine to valine. The mutant heterozygous enzyme activity decreased by about 35%, while the mutant homozygous enzyme activity decreased by about 70% (Rai & Kumar, 2021). MTHFR (1298A>C) is a mutation at nucleotide 1298 from A to C, which encodes a protein changed from glutamate to alanine. The mutant heterozygous enzyme activity decreased by about 10%, while the mutant homozygous enzyme activity decreased by about 40% (Li et al., 2022). There is a synergistic effect between the two mutations of the same enzyme, and the simultaneous mutation could further decrease the enzyme activity (Ginani et al., 2023).

Currently, most studies mainly focus on the association between the MTHFR (677C>T) polymorphism and H-type hypertension. The MTHFR (677C>T) polymorphism is significantly associated with an increased risk of H-type hypertension except in the overdominant model (Kong et al., 2022). However, few studies have explored the association between the MTHFR (677C>T) and/or MTHFR (1298A>C) and H-type hypertension. This study aims to investigate the association between the MTHFR (677C>T) and/or MTHFR (1298A>C) genotypes and H-type hypertension.

Materials and Methods

Participants

A total of 215 ischemic stroke patients with hypertension who were hospitalized in the Neuro-Brain Center of Taian City Central Hospital from June 2021 to December 2022 were enrolled. Patients were divided into two groups based on homocysteine levels: the H-type hypertension group (n = 192) with an average age of 62.6 ± 11.2 years (135 males, 57 females), and the non-H-type hypertension group (n = 23) with a mean age of 58.4 ± 11.5 years (11 males, 12 females).

Inclusion criteria: (1) Patients meeting the diagnostic criteria for hypertension in the Revised Chinese Hypertension Management Guidelines (2024 revision) (Chinese Hypertension Prevention and Treatment Guidelines Revision Committee et al., 2024), (2) Patients who underwent homocysteine testing during hospitalization, (3) Patients who underwent MTHFR gene testing during hospitalization. Exclusion Criteria: (1) Patients with thyroid disease, (2) Patients with blood system diseases (anemia, leukemia, platelet abnormalities, hemorrhagic diseases), (3) Incomplete data, (4) Patients with hepatic and renal insufficiency.

The study was approved by the Ethics Committee of the Affiliated Taian City Central Hospital of Qingdao University (No. 2021-06-50, Date: 11.05.2021). Written informed consent was obtained from the patients or their family members.

Reagents and instruments

Reagents and instruments for study: Universal sequencing reaction kits, nucleic acid purification reagents, NH4Cl, sterilized water for injection (500 mL), TL998A fluorescence detector, Eppendorf high-speed centrifuge 5,418, Eppendorf pipettes (10, 200, 1,000 μL), centrifuge (1.5 mL), pipette tips (10, 200, 1,000 μL), EDTA anticoagulation centrifuge tubes (2 mL).

Data collection

The general clinical data of the subjects were collected, including age, gender, diabetes, smoking, and drinking. Biochemical indicators were collected, including total cholesterol (TC), triglyceride (TG), low-density lipoprotein cholesterol (LDL-C), high-density lipoprotein cholesterol (HDL-C), fasting blood glucose (FBG), uric acid (UA), apolipoprotein A1 (ApoA1), apolipoprotein B (ApoB), and homocysteine concentrations (Hcy).

Specimens collection

On the day of admission or the morning of the second day, collect 1.5 mL of fasting peripheral venous blood in an EDTA anticoagulant tube, mix thoroughly, and store at 4 °C to prevent hemolysis or coagulation. The maximum storage time is 24 h.

Detecting the MTHFR gene polymorphism

(1) Add 1 mL of ammonium chloride to the centrifuge tube, then add 150 μL of venous blood and let stand for 5 min, (2) centrifuge at 3,000 rpm for 5 min, and discard the supernatant, (3) add 50 μL of nucleic acid purification reagent and mix, (4) add 1.5 μL of suspension to the corresponding universal kit for sequencing reaction. Check for any liquid residue at the front of the pipette tip and firmly fix the lid. Invert the tube several times to ensure thorough mixing, and gently tap the tube wall to remove any bubbles on the liquid surface. Use a micro centrifuge to briefly remove droplets attached to the tube wall, and test the resulting mixture using a machine according to the software number, (5) use the TL998A fluorescence detector (Xi’an Tianlong Science and Technology Co., Ltd.) for testing, (6) check the fluorescence spectrum image for gene typing.

Statistical analysis

The Hardy-Weinberg equilibrium test was used to verify whether the sample was representative of the general population. Linkage disequilibrium (LD) was analyzed using the online tool SHEsis (Shi & He, 2005). All statistical analyses were performed using SPSS 25.0. Continuous data conforming to a normal distribution were presented as mean ± standard deviation ( x¯ ± SD); continuous data conforming to a normal distribution were presented as M (P25, P75). Categorized data were represented as counts (%). Comparison between groups was performed using the chi-square test, t-test, and rank-sum test. Multivariate logistic regression analysis was used to analyze the association between MTHFR (677C>T) and/or MTHFR (1298A>C) and H-type hypertension. P < 0.05 indicates that the difference was statistically significant.

Results

Hardy-Weinberg genetic equilibrium

The Hardy-Weinberg genetic equilibrium results show that the distribution of MTHFR (677C>T) and MTHFR (1298A>C) genotype in two groups are in genetic balance (non-H-type hypertension group, χ2 = 1.415, P = 0.493 and χ2 = 0.107, P = 0.743. H-type hypertension group, χ2 = 0.838, P = 0.658 and χ2 = 1.47, P = 0.48). These results indicate that the included subjects are representative of the general population.

Characteristics of the study population

Among the 215 patients enrolled in this study, 192 patients had H-type hypertension, accounting for 89.3% of the hypertensive population. There were no statistically significant differences between the two groups in terms of age, diabetes, smoking, drinking, TG, TC, LDL-C, HDL-C, FPG, UA, ApoA1, and ApoB between the two groups (P > 0.05). However, the proportion of males in the H-type hypertension group was significantly higher than that in the non-H-type hypertension group (P < 0.05) (Table 1).

Table 1 Baseline characteristics.

Characteristics	H-type hypertension group (192)	Non-H-type hypertension group (23)	χ2/t/z	P	
Age (years)	62.6 ± 11.2	58.4 ± 11.5	1.696	0.091	
gender (n, %)	135 (70.3%)	11 (47.8%)	4.765	0.029	
Diabetes (n, %)	49 (25.5%)	5 (21.7%)	0.156	0.693	
Smoking (n, %)	67 (34.9%)	6 (26.1%)	0.711	0.399	
Drinking (n, %)	72 (37.5%)	8 (34.8%)	0.065	0.799	
TG (mmol/L)	4.46 ± 1.04	4.41 ± 1.14	0.246	0.806	
TC (mmol/L)	1.23 (0.89, 1.75)	1.56 ± 0.64	1.282	0.2	
LDL-C (mmol/L)	2.83 ± 0.84	2.83 ± 0.94	0.06	0.995	
HDL-C (mmol/L)	1.21 ± 0.28	1.18 ± 0.33	0.485	0.628	
FBG (mmol/L)	5.72 (4.94, 7.07)	5.48 (4.81, 6.34)	0.922	0.356	
UA (mmol/L)	285 (236, 336)	267 ± 79.5	0.782	0.434	
ApoA1 (g/L)	1.22 ± 0.17	1.23 ± 0.19	0.245	0.807	
ApoB (g/L)	0.96 ± 0.23	0.97 ± 0.27	0.171	0.864	
Note:

TG, Triglycerides; TC, Total cholesterol; LDL-C, Low-density lipoprotein cholesterol; HDL-C, High-density lipoprotein cholesterol; FPG, Fasting plasma glucose; UA, Uric acid; ApoA1, Apolipoprotein A1 and ApoB, Apolipoprotein B.

Distribution of MTHFR (677C>T) and MTHFR (1298A>C) genotypes and alleles

The distribution frequency of MTHFR (677C>T) CC genotype and CT genotype in the non-H-type hypertension group was significantly higher than that in the H-type hypertension group, while the distribution frequency of the TT genotype was lower than that in the H-type hypertension group, and the difference was statistically significant (P < 0.05). The distribution frequency of the MTHFR C allele in the non-H-type hypertension group was significantly higher than that in the H-type hypertension group, and the difference was statistically significant (P < 0.05). However, there were no significant differences in the genotype or allele distributions of MTHFR (1298A>C) between the two groups (Table 2).

Table 2 The distribution of MTHFR (677C>T) and MTHFR (1298A>C) genotypes and alleles.

Gene	Genetype/
allele	N (%)	H-type hypertension
group (%)	Non-H-type hypertension
group (%)	χ 2	P	
MTHFR
(677C>T)							
	CC	29 (13.5%)	25 (13.0%)	4 (17.4%)	7.664	0.022	
	CT	91 (42.3%)	76 (39.6%)	15 (65.2%)			
	TT	95 (44.2%)	91 (47.4%)	4 (17.4%)			
	C	149 (34.7%)	126 (32.8%)	23 (50.0%)	5.359	0.021	
	T	281 (65.3%)	258 (67.2%)	23 (50.0%)			
MTHFR
(1298A>C)							
	AA	164 (76.3%)	148 (77.1%)	16 (69.6%)	1.983	0.371	
	AC	45 (20.9%)	38 (19.8%)	7 (30.4%)			
	CC	6 (2.8%)	6 (3.1%)	0 (0%)			
	A	373 (86.7%)	334 (87.0%)	39 (84.8%)	0.172	0.678	
	C	57 (13.3%)	50 (13.0%)	7 (15.2%)			

Logistic regression analysis of factors of H-type hypertension

Using H-type hypertension status as a dependent variable (non-H-type hypertension = 0, H-type hypertension = 1), a logistic regression analysis was conducted with gender (female = 0, male = 1), age, MTHFR (677C>T) genotype (CC = 0, CT = 1, TT = 2), MTHFR (1298A>C) genotype (AA = 0, AC = 1, CC = 2), diabetes (no = 0, yes = 1), smoking (no = 0, yes = 1), drinking (no = 0, yes = 1), TG, TC, LDL-C, HDL-C, FBG, UA, ApoA1 and ApoB as independent variables. The results showed that gender, age, and MTHFR (677C>T) TT genotype were significantly associated with H-type hypertension (P < 0.05). Gender (P = 0.018, OR = 4.845, 95%CI [1.309–17.939]) and age (P = 0.037, OR = 1.05, 95%CI [1.003–1.100]) were identified as independent risk factors. The MTHFR (677C>T) TT genotype was an independent risk factor for H-type hypertension (P = 0.021, OR = 2.615, 95%CI [1.154–5.926]) (Table 3).

Table 3 Multivariate logistic regression analysis of the influencing factors of H-type hypertension.

Variables	Group	β	SE	Wald	P	OR	OR (95%CI)	
Gender	Female*							
		1.578	0.668	5.583	0.018	4.845	[1.309–17.939]	
Age		0.049	0.024	4.353	0.037	1.05	[1.003–1.100]	
MTHFR (677C>T)	0.961	0.417	5.302	0.021	2.615	[1.154–5.926]	
MTHFR (1298A>C)	0.307	0.529	0.338	0.561	1.36	[0.482–3.832]	
Diabetes	None*							
		0.072	0.636	0.013	0.91	1.075	[0.309–3.737]	
Smoking	None*							
		−0.124	0.695	0.032	0.858	0.883	[0.226–3.450]	
Drinking	None*							
		−0.409	0.647	0.399	0.527	0.664	[0.187–2.362]	
TC		1.985	1.68	1.396	0.237	7.277	[0.270–195.915]	
TG		−0.363	0.519	0.488	0.485	0.696	[0.251–1.925]	
LDL-C		−0.981	1.527	0.412	0.521	0.375	[0.019–7.477]	
FBG		0.013	0.109	0.014	0.906	1.013	[0.818–1.255]	
UA		0.004	0.004	1.085	0.298	1.004	[0.997–1.011]	
apoA1		−4.96	3.149	2.482	0.115	0.007	[0.000–3.357]	
apoB		−3.839	3.967	0.936	0.333	0.022	[0.000–51.280]	
HDL-C		0.397	3.094	0.016	0.898	1.487	[0.003–639.855]	
Note:

* Control group

Distribution of MTHFR (677C>T) and MTHFR (1298A>C) linked genotypes

In the H-type hypertension group, there are 7 linked genotypes of MTHFR (677C>T) and MTHFR (1298A>C), which are CC/AA (5.2%), CC/AC (4.7%), CC/CC (3.1%), CT/AA (25.5%), CT/AC (21.7%), TT/AA (46.4%), and TT/AC (1.0%). The most frequent genotype was TT/AA (46.4%) (Table 4). In the non-H-type hypertension group, there are five linked genotypes of MTHFR (677C>T) and MTHFR (1298A>C), which are CC/AA (8.7%), CC/AC (8.7%), CT/AA (43.5%), CT/AC (21.7%), TT/AA (17.4%). The highest frequency was CT/AA (43.5%) (Table 5).

Table 4 The distribution of MTHFR (677C>T) and MTHFR (1298A>C) linked genotypes in H-type hypertension group.

MTHFR (677C>T)		MTHFR (1298A>C)		
AA	AC	CC	
CC	10 (5.2%)	9 (4.7%)	6 (3.1%)	
CT	49 (25.5%)	27 (21.7%)	0 (0%)	
TT	89 (46.4%)	2 (1.0%)	0 (0%)	

Table 5 The distribution of MTHFR (677C>T) and MTHFR (1298A>C) linked genotypes in non-H-type hypertension group.

MTHFR (677C>T)		MTHFR (1298A>C)		
AA	AC	CC	
CC	2 (8.7%)	2 (8.7%)	0 (0%)	
CT	10 (43.5%)	5 (21.7%)	0 (0%)	
TT	4 (17.4%)	0 (0%)	0 (0%)	

Relationship between MTHFR (677C>T) and MTHFR (1298A>C) interaction and H-type hypertension

Linkage disequilibrium (LD) between MTHFR (677C>T) and MTHFR (1298A>C) was analyzed using SHEsis online software (http://analysis.bio-x.cn/), where D′ and r2 represent the degree of linkage disequilibrium (Ardlie, Kruglyak & Seielstad, 2002). The results showed that the 2 loci were in linkage disequilibrium (D′ = 0.933) and the degree of correlation (r2 = 0.251) (Fig. 1). For haplotypes with a frequency >3%, there were three haplotypes involved in MTHFR (677C>T) and MTHFR (1298A>C). The C-A haplotype was a protective factor for H-type hypertension (P = 0.028, OR = 0.485, 95%CI [0.252–0.934]), whereas the T-A haplotype was a risk factor (P = 0.022, OR = 2.029, 95%CI [1.096–3.756]) (Table 6).

Figure 1 The linkage disequilibrium profiles of MTHFR (677C>T) and MTHFR (1298A>C).

Table 6 The association of MTHFR gene haplotypes with H-type hypertension.

Haplotype	Frequency	OR (95%CI)	P	
H-type hypertension
group (n = 192)	Non-H-type hypertension
group (n = 23)	
C-A	0.204	0.348	0.485 [0.252–0.934]	0.028	
C-C	0.124	0.152	0.794 [0.336–1.876]	0.598	
T-A	0.666	0.5	2.029 [1.096–3.756]	0.022	
Note:

The haplotype order was MTHFR (677C>T) and MTHFR (1298A>C).

Distribution of MTHFR enzyme activity in the two groups

Based on the linked genotypes of MTHFR (677C>T) and MTHFR (1298A>C), MTHFR enzyme activity can classify into four levels: normal enzyme activity (CC/AA), mild metabolic impairment (CC/AC), moderate metabolic impairment (CC/CC, CT/AA) and severe metabolic impairment (CT/AC, CT/CC, TT/AA, TT/AC, TT/CC) (Cai et al., 2024; Deng et al., 2015). There was no significant difference in the distribution of MTHFR enzyme activity between the two groups (P > 0.05) (Table 7).

Table 7 The distribution of MTHFR enzyme activity in the two groups.

Enzyme activity	MTHFR (677C>T)/(1298A>C)	Cases number n (%)	H-type hypertension
group (%)	Non-H-type hypertension
group (%)	χ 2	P	
NEA		12 (5.6%)	10 (5.2%)	2 (8.7%)	4.321	0.229	
	CC/AA	12 (5.6%)	10 (5.2%)	2 (8.7%)			
MIMI		11 (5.1%)	9 (4.7%)	2 (8.7%)			
	CC/AC	11 (5.1%)	9 (4.7%)	2 (8.7%)			
MOMI		65 (30.2%)	55 (28.6%)	10 (43.5%)			
	CC/CC	6 (2.8%)	6 (3.1%)	0 (0%)			
	CT/AA	59 (27.4%)	49 (25.5%)	10 (43.5%)			
EVMI		127 (59.1%)	118 (61.5%)	9 (39.1%)			
	CT/AC	32 (14.9%)	27 (14.1%)	5 (21.7%)			
	CT/CC	0 (0%)	0 (0%)	0 (0%)			
	TT/AA	93 (43.3%)	89 (46.4%)	4 (17.4%)			
	TT/AC	2 (0.9%)	2 (1.0%)	0 (0%)			
	TT/CC	0 (0%)	0 (0%)	0 (0%)			
Note:

NOEA, normal enzyme activity; MIMI, mild metabolic impairment; MOMI, moderate metabolic impairment; EVMI, evere metabolic impairment.

Discussion

This study investigated the association between the genotypes of MTHFR (677C>T) and/or MTHFR (1298A>C) and H-type hypertension. Our research results indicate that the MTHFR (677C>T) TT genotype is an independent risk factor for H-type hypertension (P = 0.021, OR = 2.615, 95%CI [1.154–5.926]). Additionally, the C-A haplotype was a protective factor for H-type hypertension (P = 0.028, OR = 0.485, 95%CI [0.252–0.934]), while the T-A haplotype was a risk factor (P = 0.022, OR = 2.029, 95%CI [1.096–3.756]).

In the study, H-type hypertension patients accounted for about 89.3% of the hypertensive population, which is higher than the results of previous studies, which showed that H-type hypertension accounted for 75% of the hypertensive population in China (Chen et al., 2023). This discrepancy may be attributed to our relatively small sample size and regional population characteristics. In our study, the frequencies of the MTHFR (677C>T) wild-type homozygous CC, mutant heterozygous CT, and mutant homozygous TT genotypes are 13.5%, 42.3%, and 42.2%, respectively, with the T allele frequency of 65.3%, which was consistent with previous research in Shandong Province, where the T allele frequency was 63.1% (Fan et al., 2016). The frequencies of the MTHFR (1298A>C) wild-type homozygous AA, mutant heterozygous AC, and mutant homozygous CC genotypes are 76.3%, 20.9%, and 2.8%, respectively, with the C allele frequency of 13.3%. This was consistent with previous research results in Shandong Province, where the frequency of the C allele is 13.1% (Fan et al., 2016).

Previous studies have demonstrated that MTHFR (677C>T) polymorphism is associated with H-type hypertension in elderly individuals (P = 0.024, OR = 7.335, 95%CI [1.303–41.302]) (Zhang, Dou & Zhou, 2024). Additionally, the MTHFR (677C>T) allele mutation can increase the risk of ischemic stroke in older adults (Chang et al., 2019). In the codominant and recessive models, the MTHFR (677C>T) TT genotype was the strongest determinant of H-type hypertension (Ma et al., 2022). Our research results indicate that the MTHFR (677C>T) TT genotype was an independent risk factor for H-type hypertension (P = 0.021, OR = 2.615, 95%CI [1.154–5.926]), which was consistent with the results of previous studies. Currently, the results of studies on the association between the MTHFR (1298A>C) gene polymorphism and hypertension are still inconsistent (Liu et al., 2019; Mabhida et al., 2022; Yuan et al., 2019; Zhang et al., 2023b). These differences are likely due to ethnic and regional variations in study populations. Consistent with some of these reports, our study found no significant association between the MTHFR (1298A>C) polymorphism and H-type hypertension.

The results of linkage and haplotype analysis of MTHFR (677C>T) and MTHFR (1298A>C) show that there were seven kinds of haplotype combinations constructed by the 2 loci in the H-type hypertension group. CT/CC and TT/CC were not detected, with TT/AA being the most common (46.4%). In the non-H-type hypertension group, there were five haplotype combinations constructed by the 2 loci, CC/CC, CT/CC, TT/AC and TT/CC were absent, and the highest frequency was CT/AA (43.5%). These distributions are broadly consistent with prior findings (Fan et al., 2016). The results of our study showed that the MTHFR (677C>T) and MTHFR (1298A>C) were in linkage disequilibrium (D′ = 0.933, r2= 0.251), aligning with earlier observations of LD between these loci (Ginani et al., 2023). For haplotypes with a frequency >3%, there were three haplotypes in MTHFR (677C>T) and MTHFR (1298A>C). The C-A haplotype was a protective factor for H-type hypertension (P = 0.028, OR = 0.485, 95%CI [0.252–0.934]), while the T-A haplotype was a risk factor (P = 0.022, OR = 2.029, 95%CI [1.096–3.756]).

This study has certain limitations: Firstly, the sample size was small, and no genotypes with low mutation rates were not detected. Secondly, this study was a retrospective analysis. Thirdly, this study did not include healthy individuals for comparison. Future research will expand the sample size, conduct prospective cohort studies, and include healthy individuals as controls to enhance the reliability and persuasiveness of the results, thereby providing a scientific basis for personalized diagnosis and treatment.

Conclusion

In patients with ischemic stroke, the MTHFR (677C>T) TT genotype was an independent risk factor for H-type hypertension. For haplotypes with a frequency >3%, there were three haplotypes in MTHFR (677C>T) and MTHFR (1298A>C). The C-A haplotype was a protective factor for H-type hypertension, while the T-A haplotype was a risk factor.

In the future, we will expand the sample size, adopt a prospective cohort design, and include healthy individuals as controls to clarify the specific association between MTHFR (677C>T), MTHFR (1298A>C) polymorphisms and related indicators and ischemic stroke. We will also explore individualized interventions for different genotypes, evaluate their effects, and provide scientific evidence for risk stratification and precision prevention and treatment of the population.

Supplemental Information

Supplemental Information 1 Dataset.

Grammarly was used for language editing and improving manuscript clarity.

Additional Information and Declarations

Competing Interests

The authors declare that they have no competing interests.

Author Contributions

Bo Zhou conceived and designed the experiments, performed the experiments, analyzed the data, authored or reviewed drafts of the article, and approved the final draft.

Tingting Yang analyzed the data, prepared figures and/or tables, and approved the final draft.

Shicang An performed the experiments, analyzed the data, prepared figures and/or tables, and approved the final draft.

Qike Xu performed the experiments, prepared figures and/or tables, and approved the final draft.

Yuna Liang conceived and designed the experiments, authored or reviewed drafts of the article, and approved the final draft.

Xiangyang An conceived and designed the experiments, prepared figures and/or tables, authored or reviewed drafts of the article, and approved the final draft.

Human Ethics

The following information was supplied relating to ethical approvals (i.e., approving body and any reference numbers):

The study was approved by the Affiliated Taian City Central Hospital of Qingdao University Ethics Committee of the institute (No. 2021-06-50 date: 11.05.2021).

Data Availability

The following information was supplied regarding data availability:

The raw data is available in the Supplemental File.

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
