# Peer review of "Association between MTHFR gene polymorphisms and H-type hypertension in patients with ischemic stroke"

_PeerJ, doi:10.7717/peerj.20210_

## Round 0.1 · original submission · Major Revisions

**Language Note:** The review process has identified that the English language must be improved. PeerJ can provide language editing services - please contact us at [email protected] for pricing (be sure to provide your manuscript number and title). Alternatively, you should make your own arrangements to improve the language quality and provide details in your response letter. – PeerJ Staff

Reviewer 1 ·

Basic reporting

1. Abstract: In the Background section of the abstract, the current phrasing—"To discuss the association between methylenetetrahydrofolate reductase (MTHFR) gene polymorphisms and ischemic stroke in patients with H-type hypertension"—may cause ambiguity. A more precise and appropriate wording might be: "To discuss the association between methylenetetrahydrofolate reductase (MTHFR) gene polymorphisms and H-type hypertension in ischemic stroke patients." This revision would more clearly reflect the study’s focus.

2. Introduction:
- The introduction provides a broad overview of the topic; however, there are a few areas that require improvement in grammar and clarity:
1. Sentence structure and clarity: Lines 45–47 contain awkward phrasing. Consider revising for clearer sentence flow and logical coherence.
2. Article usage and noun agreement: Line 56–57 includes the phrase "a important enzymes", which should be corrected to "an important enzyme" to ensure grammatical accuracy.
3. Verb tense consistency: For example, in lines 62–63, the verb tense is incorrect. Please revise “which lead the encoded protein...” to “which leads to a change in the encoded protein...”.

- Scientific accuracy should also be ensured:
1. In line 65–66, the phrase "There was a synergistic effect between the two enzymes" is inaccurate, as it refers to two mutations of the same enzyme, not two separate enzymes. Please revise to "There is a synergistic effect between the two mutations of the same enzyme."

- For international readability, it is recommended to:
1. Replace the phrase "our country" (line 48) with the specific country name (e.g., "China") for clarity and relevance to a global audience.
2. Clarify terminology by specifying the types of hypertension referred to in line 52.
3. Clearly indicate the type(s) of stroke involved in line 53—whether the study focuses on ischemic stroke, hemorrhagic stroke, or both.

3. Literature references, figures, and tables provided are already sufficient

Experimental design

1. The research question is clearly defined and relevant.
2. Please provide further detail regarding the exclusion criterion "patients with blood system diseases." For clarity and reproducibility, consider listing specific diseases or conditions under this category.
3. The methodology, including data collection and statistical analysis, is well described and sufficiently detailed to allow replication.

Validity of the findings

1. The findings are statistically sound
2. Conclusion: it is suggested to re-emphasize the study population by including "ischemic stroke patients" to reinforce the context for the reader.

Additional comments

- It is recommended to expand on the discussion by including the clinical context and practical implications of the findings, if applicable.
- Please state the limitations of the study and suggest directions for future research to strengthen the discussion section.

Reviewer 2 ·

Basic reporting

Title & Abstract
Title:
The title is adequate as it identifies the population under study (patients with ischemic stroke) and the conditions of interest (MTHFR gene polymorphisms and H-type hypertension).
Abstract:
The abstract is structured according to the journal requirements. However, the subheadings should be bolded. In the background, more details about the reasons leading to the conduction of the study should be provided. The objective is not adequate, as the real aim of the study was to assess the association of two MTHFR polymorphisms on increased homocysteine levels in patients with hypertension and stroke. The methods section should be improved. Essential inclusion/exclusion criteria should be provided. Authors should also specify which data was collected and the methods used for all measurements. The sequencing method and databases used for the interpretation of results should also be specified. In the results section, the percentage of patients with elevated homocysteine levels over those with hypertension and stroke is given. As will be discussed later this is not in line with the study’s objectives. Please provide numbers to describe variables, including the proportion of males and the distribution of MTHFR polymorphisms, among others. Please specify which is the risk (or minor) allele for each polymorphism. Please specify which variables were included in the multivariate model.

Introduction
The introduction outlines the importance hypertension and H-type hypertension for public in China. While the focus of this study is on stroke patients, its relationship with H-type hypertension is only briefly mentioned (Page 6 Lines 52-55). Please give more details about the relationship between these conditions. Please review the following references:
- Chen, K., He, J., Fu, L. et al. Prediction of ischemic stroke in patients with H-type hypertension based on biomarker. Sci Rep 15, 1221 (2025). https://doi.org/10.1038/s41598-024-83662-3
- Zhao, M., Wang, X., He, M., Qin, X., Tang, G., Huo, Y., ... & Cai, Y. (2017). Homocysteine and stroke risk: modifying effect of methylenetetrahydrofolate reductase C677T polymorphism and folic acid intervention. Stroke, 48(5), 1183-1190.
Please provide a reference to support this statement: “MTHFR(677C>T) and MTHFR(1298A>C) were the common mutation sites of MTHFR, and they have synergistic effect.” (Page 6 Lines 57-8)
Authors state that “most studies mainly focus on the association between MTHFR(677C>T) and H-type hypertension” (Page 6 Lines 67-8). However, this association should be further discussed. Please review the following study:
- Kong Y, Zheng J, Li L, Lu L, Wang J. Association of MTHFR Polymorphisms with H-Type Hypertension: A Systemic Review and Network Meta-Analysis of Diagnostic Test Accuracy. Int J Hypertens. 2022 Mar 22; 2022:2861444.
The objective should be: “to investigate the association between the MTHFR(677C>T) and MTHFR(1298A>C) polymorphisms and increased homocysteine levels in patients with stroke and hypertension.

Figures & Tables
All tables and figures are cited in the text and contains appropriate headers and footers.

Experimental design

Material and Methods
The inclusion and exclusion criteria are not adequate (Page 7 Lines 74-83). Only patients with homocysteine level was g10µmol/L, which means that most patients will have H-type hypertension, thus biasing results. Furthermore, only those who underwent MTHFR gene testing during hospitalization were selected, which produces a selection bias that affects negatively the quality of the study results.
Please specify how were polymorphism assessed: type of sequencing used, if a chip was used, and the databases used to detect the polymorphisms (Page 8 Line 111). Please also specify which are the minor (risk) alleles for each polymorphism. The statistical analysis is correct.

Validity of the findings

Results
The results showed that the HW equilibrium was reached. Table 1 compares the characteristics of H-type and non-H-type hypertensive. The proportion of H-type hypertension is higher than expected, which is related to the inclusion criteria set, as discussed earlier. The frequency of MTHFR polymorphisms in these groups of patients is compared in Table 2. Differences were only noted in the (677C>T) variant. The independent effects of this variant are confirmed by the logistic regression analysis in Table 3. Finally, Tables 4-7 and Figure 1 evaluated a linkage disequilibrium between the variants. The results are clearly presented.

Discussion
This study confirms the association between MTHFR (677C>T) variant with h-type hypertension in patients with stroke and hypertension. These results were expected based on the available evidence, and have been previously reported (Int J Clin Exp Med 2020;13(12):9892-9897).
The first two paragraphs of the discussion summarize the findings (Page 10 Lines 179-195). In Lines 185-189 please also mention that the selection criteria may account for the high prevalence of h-type HT. The linkage disequilibrium analysis is discussed later (Page 11 Lines 206-217).
The discussion fails to mention the limitations of the study and the importance of the findings for clinical care of stroke h-type patients.

Conclusion
The conclusion adequately states the findings. Future lines or research and impact for clinical care should be proposed.

---

## Round 0.2 · accepted · Accept

Thank you for revising your manuscript to address the reviewers' concerns. Reviewer 2 recommends acceptance with some minor modifications to the text and I am satisfied with your response to the earlier comments of reviewer 1. The manuscript will be ready for publication when you have addressed the remaining comments of reviewer 2.

Reviewer 2 ·

Basic reporting

Title & Abstract
The title is adequate as it identifies the population under study (patients with ischemic stroke) and the conditions of interest (MTHFR gene polymorphisms and H-type hypertension).
The abstract has improved significantly. It complies with journal’s requirements and adequately reflects the methodology applied to the study and its main results. The conclusion is clearly expressed and does not overstate findings.

Introduction
The authors now discuss these references as suggested. The reference is adequate which is added to support this statement: “MTHFR (677C>T) and MTHFR (1298A>C) were the common mutation sites of MTHFR, and they have synergistic effect.” (Page 6 Lines 57-8).
The modifications are adequate, contributing to the increased quality of the manuscript.
The objective is still not adequate. It should reflect that only hypertension in stroke patients is targeted. Please correct.

Figures & Tables
All tables and figures are cited in the text and contain appropriate headers and footers.

Experimental design

Material and Methods
The changes are adequate and reduces the likelihood of selection bias. Author’s response is sound. This should be cited as a limitation in the discussion section. The methodology employed stating “Using the TL998A fluorescence detector (Xi’an Tianlong Science and Technology Co., Ltd.) for testing” is adequate.

Validity of the findings

Results
This section is sound and no revisions are required.

Discussion
The authors adequately describes the study limitations. Please also note that only patients in whom the gene was sequenced were included, as previously discussed.

Conclusion
The additions done in the conclusions section are sound.